# The Effect of Necrosis Inhibitor on Dextran Sulfate Sodium Induced Chronic Colitis Model in Mice

**DOI:** 10.3390/pharmaceutics15010222

**Published:** 2023-01-09

**Authors:** Dongwoo Kim, Ja Seol Koo, Soon Ha Kim, Yeong Seo Park, Jung Wan Choe, Seung Young Kim, Jong Jin Hyun, Sung Woo Jung, Young Kul Jung, Hyung Joon Yim

**Affiliations:** 1Division of Gastroenterology and Hepatology, Department of Internal Medicine, Korea University Ansan Hospital, Korea University College of Medicine, Seoul 02841, Republic of Korea; 2MitoImmune Therapeutics Inc., Seoul 06123, Republic of Korea

**Keywords:** necrosis inhibitor, chronic colitis, inflammatory bowel disease

## Abstract

Uncontrolled chronic inflammation and necrosis is characteristic of inflammatory bowel disease (IBD). This study aimed to investigate the effect of necrosis inhibitor (NI, NecroX-7) on a dextran sulfate sodium (DSS) induced chronic colitis model of mice. DSS was administered on days 1–5, and the NI was administered intraperitoneally (3 mg/kg, 30 mg/kg) on days 1, 3, and 5 as well as every other day during the first five days of a three-week cycle. Three cycles of administration were performed. Colitis was evaluated based on the disease activity index (DAI) score, colon length, and histological score. Reverse transcription polymerase chain reaction testing, the Western blot assay, and immunohistochemical staining were performed to determine inflammatory cytokine levels. The NI reduced body weight change and the DAI score. Colon length and the histological score were longer and lower in the NI-treated groups, respectively. The NI decreased the expression of pro-inflammatory cytokines, particularly in tumor necrosis factor alpha (TNF-α) and phosphorylated nuclear factor kappa B (p-NF-κB). Immunohistochemical staining revealed decreased inducible nitric oxide synthase (iNOS) and high mobility group box 1 (HMGB1) levels. Overall, the NI improved DSS induced chronic colitis by attenuating the mRNA expression of pro-inflammatory cytokines such as TNF-α. Therefore, NI use is a potential, novel treatment approach for IBD.

## 1. Introduction

Inflammatory bowel disease (IBD) is a chronic inflammatory disease affecting the gastrointestinal tract. In patients with IBD, chronic and persistent inflammation develops in the intestinal epithelium and is self-limiting, unlike most inflammatory responses [1]. This is presumably related to a defective immune system [2]. The conventional treatment for IBD, such as 5-aminosalycylic acid, steroids, and azathioprine, focuses on reducing the inflammatory response [3]. Several biologics, such as the anti-tumor necrosis factor alpha (TNF- α) antibody, anti-α4β7 antibody, interleukin (IL) 12/23 antibody, and Janus kinase inhibitor, have recently been developed and used in IBD treatment. However, these drugs’ therapeutic effects are limited and they are merely effective in approximately 50% of patients with IBD [4,5,6,7].

Several forms of cell death exist, including apoptosis and necrosis, which are considered two major distinct processes. Apoptosis, the predominant form of physiological cell death, may be initiated by cell damage or interaction with immune system [8]. Apoptosis is a process resulting in cell death without spillage of cell contents into the surrounding environment, therefore, it does not cause inflammatory responses [9]. In contrast, necrosis is uncontrolled cell death caused by external damage, such as inflammation or hypoxia. This process is characterized by cell-membrane destruction and intracellular material discharge due to pro-inflammatory substances, causing inflammatory reactions [8]. Previous studies have reported increased apoptosis in patients with IBD, especially those with ulcerative colitis [10,11]. However, other studies have reported an association between necrosis and IBD [12,13,14]. Therefore, these results suggest that suppression of necrosis may be a new IBD-treatment method. However, few studies have investigated necrosis inhibition in IBD due to the unavailability of drugs that can regulate necrosis.

NecroX-7 (MitoImmune Therapeutics, Seoul, Republic of Korea) is an indole-derived chemical compound, composing C_25_H_32_N_4_O_4_S_2_ and was recently developed as a necrosis inhibitor (NI). NIs inhibit cell necrosis by blocking calcium intake into mitochondria and acts as a free radical scavenger, with minimal effects on apoptosis [15,16]. The effects of NIs have been reported in various organs. Nis have been found to attenuate acute graft-versus-host disease [17], prevent oxidative stress in cardiomyopathy [18], and to improve renal ischemia-reperfusion injury [19]. Their anti-inflammatory effects in an acute colitis model have also been reported. In our preliminary study, NIs reduced necrotic cell death and ameliorated dextran sulfate sodium (DSS)-induced acute colitis [20].

This study aimed to evaluate the effect of NIs in a DSS induced chronic colitis model and investigate the mechanism underlying their effect.

## 2. Materials and Methods

### 2.1. Animals and Ethics Statement

C57BL/6 mice were used in the DSS induced chronic colitis model. All animal experimental protocols were approved by the animal institutional review board of Korea University Ansan Hospital (KOREA-2019-0106) and all experiments were performed in accordance with relevant guidelines 0193.

### 2.2. Dextran Sulfate Sodium Induced Chronic Colitis Model

Eight-week-old male C57BL/6 mice were randomly assigned to six groups: (1) control group—treated exclusively with saline (*n* = 5), (2) NI3 group—treated with NI with 3 mg/kg (*n* = 5), (3) NI30 group—treated with NI 30 mg/kg (*n* = 5), (4) DSS group—treated with DSS (*n* = 12), (5) DSS + NI3 group—treated with DSS and NI 3 mg/kg (*n* = 12), (6) DSS + NI30 group—treated with DSS and NI 30 mg/kg (*n* = 12). Chronic colitis was induced using 2.5% DSS solution (25 g of DSS to 1 L of tap water). In our previous study on the effect of NIs on an acute colitis model, 30 mg/kg of NI exhibited anti-inflammatory effects [20]. Therefore, we administered 30 mg/kg of NI to the experiment group. To confirm the minimum effective dose, 3 mg/kg of NI was administered to another experimental group. One cycle of administration lasted 3 weeks. DSS was orally administered for 5 days in first week of the administration cycle, followed by the consumption of tap water for the remaining 16 days. The NI was administered via intraperitoneal injection on days 1, 3, and 5 of the first week. The initial NI dose was administered 4 h before DSS administration. This 3-week cycle was repeated three times, and mice were euthanized using CO_2_ gas on day 53, as shown in Figure 1.

### 2.3. Disease Activity Index (DAI)

Colitis severity was scored using the DAI score, which is based on body weight change, stool consistency, and bleeding. The DAI was independently assessed daily for the first 4 weeks and subsequently every other day until the end of the experiment by two researchers. Table 1 shows each parameter’s score.

### 2.4. Histological Evaluations

On day 53, mice colons were harvested, and their lengths were measured after washing. Colon tissues for polymerase chain reaction (PCR) testing and Western blot assay were saved on distal side of the colon, and the rest of the tissues were rolled using the Swiss roll technique and fixed in 10% formalin to prepare a paraffin block. The sections were stained with hematoxylin and eosin. Histological evaluation was performed for the proximal, mid, and distal segments in a blind fashion by two independent researchers and the sum of the scores was compared. The histological scores are listed in Table 2.

### 2.5. Evaluation of Pro-Inflammatory Cytokines Using Real-Time PCR

The proximal and distal segments of harvested colons were washed with phosphate buffered saline (PBS) and frozen using −78 °C liquid nitrogen. All isolated colon tissues were finely homogenized using the Tissue Lyser (QIAGEN, Hilden, Germany) and total RNA was extracted using the TRIzol^TM^ reagent (Sigma-Aldrich, St. Louis, MO, USA). The RNA concentration of each group was measured using Take3^TM^ Trio Micro-Volume Plates (BioTek, Winooski, VT, USA) and cDNA was synthesized using a reverse transcription system (iNtRON BIO, Seongnam, Republic of Korea) with 1000 ng of total RNA. The cDNA was detected using the LightCycler^®^ FastStart DNA Master SYBR Green I kit (Roche, Basel, Switzerland), and real-time PCR was performed using the Light-Cycler^®^ 480 system (Roche, Basel, Switzerland). HMGB1, TNF-α, IL-6, and beta-actin primers were purchased from Bioneer (Daejeon, Republic of Korea. Appendix A). Specific gene expression was quantified by measuring the threshold cycles (Ct values) of specific genes and comparing them to those of beta-actin.

### 2.6. Western Blot Assay

Homogenized colon tissues were dissolved in lysis buffer (protein extraction solution, NP 40; ELPIS BIOTECH, Daejeon, Republic of Korea) containing protease inhibitor (Thermo Scientific^TM^ Halt Protease and Phosphatase inhibitor Cocktail). Tissue lysate was quantified using Pierce^TM^ BCA Protein Assay Kit (Thermo Scientific^TM^, Waltham, MA, USA). Samples were separated using sodium dodecyl sulphate-polyacrylamide gel electrophoresis (SDS-PAGE) after heating for 5 min at 95 °C. After transferring the protein separated on SDS-PAGE to the polyvinylidene difluoride (PVDF) membrane, the immunoblot assay was performed. Nuclear factor kappa-light-chain enhancer of activated B cells (NF-κB), phosphor-NF-κB (p-NF-κB), and inducible nitric oxide synthase (iNOS) antibodies were used as 1′ antibodies (Appendix A). The antibodies were diluted to 1:5000. The 1′ antibodies were reacted for overnight and washed with 1× PBS. Glyceraldehyde 3-phosphate dehydrogenase (GAPDH) was measured as a loading control. Anti-mouse or anti-rabbit IgG horseradish peroxidase (BETHYL-A90-11P, A120-101P) was antibodies were diluted to 1:5000 and used as 2′ antibodies. The 2′ antibodies were reacted for 1 h and washed with PBS. Protein expression was measured using a chemiluminescence kit (Promega W1001, Seoul, Republic of Korea). All assay images were obtained using Image Lab Software (Bio-Rad Laboratories, Inc., Seoul, Republic of Korea).

### 2.7. Immunohistochemistry

Immunohistochemical staining was performed using the Avidin–Biotin Complex method (Liquid DAB Substrate Kit, Invitrogen, Waltham, MA, USA). The cross section of the paraffinized colon was sliced to 4–6 μm thickness and mounted on a glass slide (KCFC, Yongin, Republic of Korea). The slides were dried in an oven at 60 °C for 30 min. Paraffin was removed using xylene and ethanol, and rehydration was performed. For antigen retrieval, tissues were heated to 95 °C in a sodium citrate buffer (pH 6.0; ab93678, Cambridge, UK) for 20 min and 0.5% H_2_O_2_ and methanol for 30 min to inactivate endogenous peroxidase. The 1′ antigodies (HMGB1- Cell signaling Technology, iNOS-BD transduction) were diluted to 1:200 and reacted overnight. Subsequently, the 2′ antibodies (ab859043; Invitrogen, Waltham, MA, USA) were reacted after PBS washing.

### 2.8. Statistical Analysis

All data are expressed as the mean ± standard deviation and compared using the Mann–Whitney and Kruskal–Wallis tests. All statistical tests were two-tailed, and statistical significance was set at *p* < 0.05. All analyses were performed using SPSS software (version 23.0; SPSS Inc., Chicago, IL, USA).

## 3. Results

### 3.1. NI Effect on the DAI

In the DSS-treated groups (DSS, DSS + NI3, and DSS + NI30), the body weight of the mice decreased from 5 days after DSS administration (Figure 2). On day 53, the mean body weight of the DSS group decreased to 87.5% of that at baseline, while that of the DSS + NI3 group merely decreased to 95.3%, exhibiting a significantly smaller weight loss than the DSS group (*p* = 0.047). In the DSS + NI30 group, the mean body weight decreased to 96.9%; however, the decrease was not significant (*p* = 0.091). The DAI score on day 53 was significantly lower in the DSS + NI3 and DSS + NI30 groups than in the DSS group (DSS + NI3 group vs. DSS group; 5.09 vs. 8.74, *p* = 0.020; DSS + NI30 group vs. DSS group; 5.58 vs. 8.74, *p* = 0.039) (Figure 2).

### 3.2. Colon Length and Histological Score

On day 53, the lengths of the harvested colons were measured, and their mean lengths were 88.6 ± 4.0, 88.0 ± 6.4, and 81.2 ± 5.9 mm in the control, NI3, and NI30 groups, respectively. The mean colon length was 49.8 ± 5.0 mm in the DSS group. Both the DSS + NI3 (67.3 ± 8.1 mm, *p* < 0.001) and DSS + NI30 (73.3 ± 9.9 mm, *p* = 0.001) groups had significantly longer colon lengths than the DSS group (Figure 3A,B). On histological evaluation, the DSS group demonstrated increased of neutrophil and lymphocyte infiltration with severe crypt damage. In the DSS + NI3 and DSS + NI30 groups, neutrophil infiltration was less severe and crypt structure was relatively conserved. The DSS + NI3 and DSS + NI30 groups’ histological scores were also significantly lower than those of the DSS group (DSS + NI3 group vs. DSS group: 24.9 ± 6.0 vs. 33.4 ± 4.9, *p* = 0.014; DSS + NI30 group vs. DSS group: 23.0 ± 7.1 vs. 33.4 ± 4.9, *p* = 0.012) (Figure 3C,D).

### 3.3. Pro-Inflammatory Cytokine Quantitative Analysis Using Real-Time PCR

Each group’s colon tissues were quantitatively analyzed for pro-inflammatory cytokines using real-time PCR (Figure 4). In the DSS group, the levels of pro-inflammatory cytokines, such as high mobility group box 1 (HMGB1), TNF-α, and IL-6, were significantly elevated. NI treatment reduced TNF-α levels significantly (DSS + NI3 group vs. DSS group: 190.6 ± 174.3 vs. 453.0 ± 199.0, *p* = 0.013; DSS + NI30 group vs. DSS group: 136.8 ± 131.6 vs. 453.0 ± 199.0, *p* = 0.003). HMGB1 expression tended to decrease in the NI-treated groups, although the decrease was not significant (DSS + NI3 group vs. DSS group: 68.3 ± 16.0 vs. 162.2 ± 121.5, *p* = 0.151; DSS + NI30 group vs. DSS group: 80.6 ± 33.1 vs. 162.2 ± 121.5, *p* = 0.093). IL-6 expression also tended to decline in the NI-treated group (DSS + NI3 group vs. DSS group: 257.6 ± 247.8 vs. 702.0 ± 628.7, *p* = 0.067; DSS + NI30 group vs. DSS group: 308.8 ± 328.1 vs. 702.0 ± 628.7, *p* = 0.118).

### 3.4. Western Blot Assay

Figure 5 shows the NF-κB, p-NF-κB, and iNOS levels based on the western blot assay. No significant differences in NF-κB were noted among the groups (*p* = 0.142). However, elevated p-NF-κB levels were observed in the DSS-treated group, whereas significantly reduced levels were noted in the NI-treated groups (DSS + NI3 group vs. DSS group: 2.08 ± 0.53 vs. 3.05 ± 0.38, *p* = 0.021; DSS + NI30 group vs. DSS group: 2.07 ± 0.05 vs. 3.05 ± 0.38, *p* = 0.034). The NI also decreased iNOS levels significantly in the NI-treated groups compared with those in the DSS group (DSS + NI3 group vs. DSS group: 1.29 ± 0.10 vs. 1.82 ± 0.09, *p* = 0.016; DSS + NI30 group vs. DSS group: 1.08 ± 0.08 vs. 1.82 ± 0.09, *p* = 0.033) (Figure 5).

### 3.5. Immunohistochemical Staining

To confirm the anti-inflammatory effects of the NI, immunohistochemical staining was performed. The control groups exhibited no HMGB1- and iNOS-positive cells. No HBGB1- or iNOS-positive cells were observed in the NI3 and NI30 groups. In the DSS group, neutrophil infiltration and crypt damage were observed, and HMGB1- and iNOS-positive cells were clustered around the inflammatory lesions. In the DSS + NI3 and DSS + NI30 groups, HMGB1- and iNOS-positive cells were still observed. However, their counts were significantly reduced compared with those in the DSS group (Figure 6).

## 4. Discussion

The DSS induced chronic colitis model is one of the most widely used well-established animal models in IBD studies. Although the exact mechanism by which DSS causes intestinal inflammation is unknown, DSS is perceived to directly cause damage to the intestinal epithelial cells and alter the permeability of intestinal substances and bacteria via Toll-like receptors (TLRs) and the NF-κB pathway [21,22]. This promotes the secretion of inflammatory cytokines, such as TNF-α, and initiates the inflammatory response [23]. Inflammatory cells cause cell damage and necrosis by modifying intracellular calcium regulation and releasing free radicals [24,25]. NIs inhibit necrosis by blocking mitochondrial calcium intake and reducing reactive oxygen species (ROS) by acting as strong free radical scavengers [15,18,19,26]. In our preliminary in vitro study, the NI was found to mitigate necrotic cell death in the intestinal epithelial cell line (IEC-18) [20].

In the present study, the NI effectively suppressed the inflammation in the DSS induced chronic colitis model. Body weight, the DAI score, colon length and the histological score exhibited significant differences in the NI-treated groups compared with those in the DSS group. Although weight-loss reduction was subtle, the DAI score significantly differed. During the experiment, four mice died in the DSS group, and only one died in the DSS + NI30 group. No deaths occurred in theDSS + NI3 group. The decreased number of mice might have justified the subtle difference. In addition, since the mice with severe colitis died before the end of the experiment, mice with relatively mild colitis were possibly included, thus reducing the difference.

Previous studies have reported increased TNF-α and IL-6 expression in patients with IBD. IL-6 is one of the major pro-inflammatory cytokines in IBD, and it activates “signal transducer and activator of transcription 3 (STAT3)” by binding to membrane-bound receptors [27,28]. TNF-α is also an important TLR and NF-κB pathway cytokine. NF-κB phosphorylation potentially causes an inflammatory reaction [29,30]. In our study, TNF-α expression was significantly decreased, and IL-6 reduction was not significant. However, IL-6 levels tended to decrease. The small numbers of mice and large deviation of the expression value might have affected the results. The Western blot assay revealed reduced p-NF-κB levels in the NI-treated group. Phosphorylation is required for NF-κB to have transcriptional activity [31,32,33]. Although NF-κB yields no difference, reduced p-NF-κB levels play a role in the effective inhibition of inflammation by NIs.

HMGB1 is a nonhistone protein that binds to DNA in the nucleus and is a potent pro-inflammatory mediator [34,35]. Because HMGB1 is released by necrotic or damaged cells, it is potentially useful as a necrosis indicator. Several studies have also reported HMGB1 inhibition and pro-inflammatory cytokine reduction by NIs. Jin et al. demonstrated that Nis attenuated HMGB1 and reduced TNF-α significantly [19]. Im et al. reported that NIs inhibited the release of TNF-α and IL-6 associated with HMGB1 decease in graft-versus-host disease [17]. In oral mucositis, Im et al. revealed that NIs inhibited HMGB1 and the HMGB-1-induced release of TNF-α and NF-κB, exhibiting consistency with our study [36]. These studies all indicated that NIs reduce inflammation via HMGB1 reduction. In this study, HMGB1 expression in real-time PCR appeared to be decreased in the NI-treated groups, though the decrease was not statistically significant. As previously mentioned, this might have been due to the small number of mice. Decreased HMGB1-positive cells on immunohistochemical staining potentially supports HMGB1 inhibition by NIs. Moreover, HMGB1 reportedly possesses three subtypes [37], and the degree of its release from inflammatory and necrotic cells varies [38,39,40]. Several studies have suggested that HMGB1 also serves a role in tissue repair and healing [41,42,43]. Redox forms of HMGB1 are considered to be associated with regeneration. Therefore, in the late phase of chronic inflammation, the redox form of HMGB1 may abound and plays a role associated with tissue repair, possibly justifying the insignificant decrease in HMGB1. Although we did not measure HMGB1 subtypes separately, measuring them may help clarify the relationship between HMGB1 and the anti-inflammatory effect of NIs.

NIs are also well-established, strong free radical scavengers. They potentially inhibit inflammation by reducing ROS [44]. NO produced by iNOS has a potential role as a proinflammatory mediator in IBD. ROS are generated during an inflammatory response and occasionally activate NF-κB via the cross reaction [45]. The exact mechanism underlying NO-induced cellular damage remains unknown, though NO is believed to interact with oxygen radicals. Recent studies have confirmed that iNOS expression correlates with C-reactive protein levels and the severity of inflammation [46,47]. In our study, iNOS levels were significantly reduced in the NI-treated groups in both Western blotting and immunohistochemical staining compared with those in the DSS group. NIs can suppress inflammatory reactions directly. However, they may also contribute to an attenuated inflammatory response by alleviating oxygen stress.

In summary, our findings demonstrate that NIs improve DSS induced chronic colitis and reduce the expression of pro-inflammatory cytokines, such as TNF-α. Therefore, inhibition of necrosis is a potential, novel therapeutic option for the IBD treatment.

## Figures and Tables

**Figure 1 pharmaceutics-15-00222-f001:**
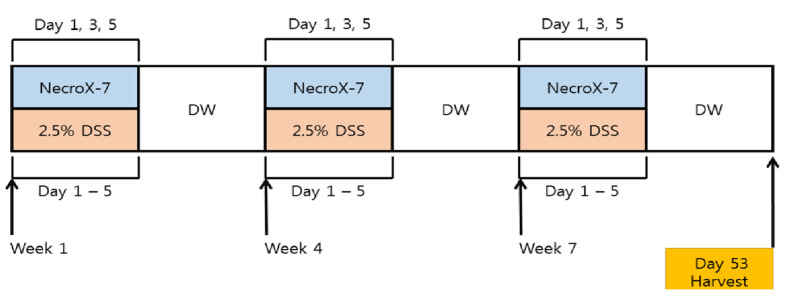
Schematic protocols of DSS induced chronic colitis model in mice. (Abbreviations) DSS, dextran sulfate sodium.

**Figure 2 pharmaceutics-15-00222-f002:**
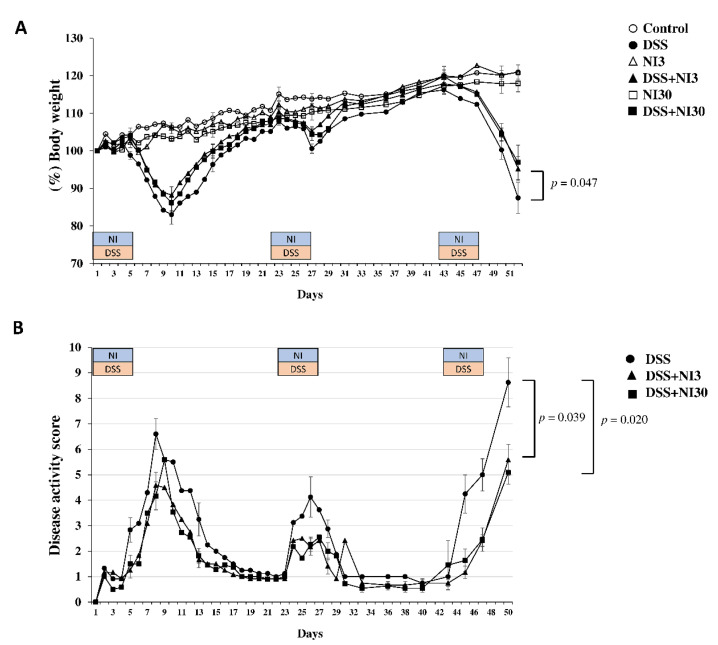
(**A**) Change in body weight of mice. (**B**) Change in disease activity index score. (Abbreviations) DSS, dextran sulfate sodium; NI, necrosis inhibitor.

**Figure 3 pharmaceutics-15-00222-f003:**
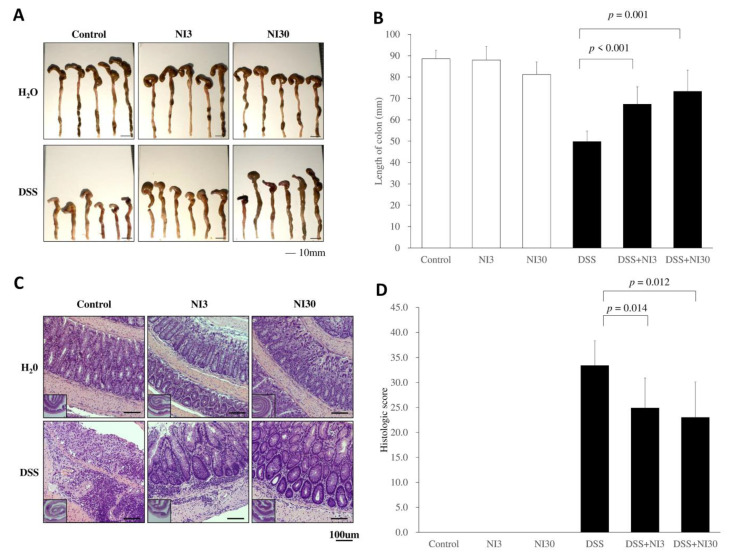
Effect of the necrosis inhibitor on the colon length and histological score of each DSS induced chronic colitis group. (**A**) Gross colon morphology. (**B**) Colon length was longer in the NI-treated groups than DSS group. (**C**) Histopathology of colon (hematoxylin and eosin stain, original magnification 100×). (**D**) Histologic score is decreased in NI-treated groups than in the DSS group. (Abbreviations) DSS, dextran sulfate sodium; NI, necrosis inhibitor.

**Figure 4 pharmaceutics-15-00222-f004:**
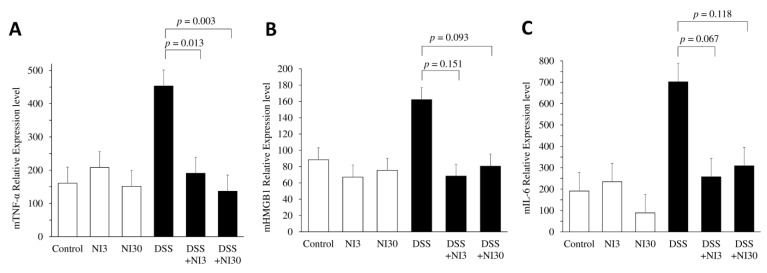
The necrosis inhibitor reduced the mRNA expression levels of pro-inflammatory cytokines. (**A**) TNF-α, (**B**) HMGB1, and (**C**) IL-6 mRNA expression levels as analyzed using real time PCR. (Abbreviations) DSS, dextran sulfate sodium; NI, necrosis inhibitor.

**Figure 5 pharmaceutics-15-00222-f005:**
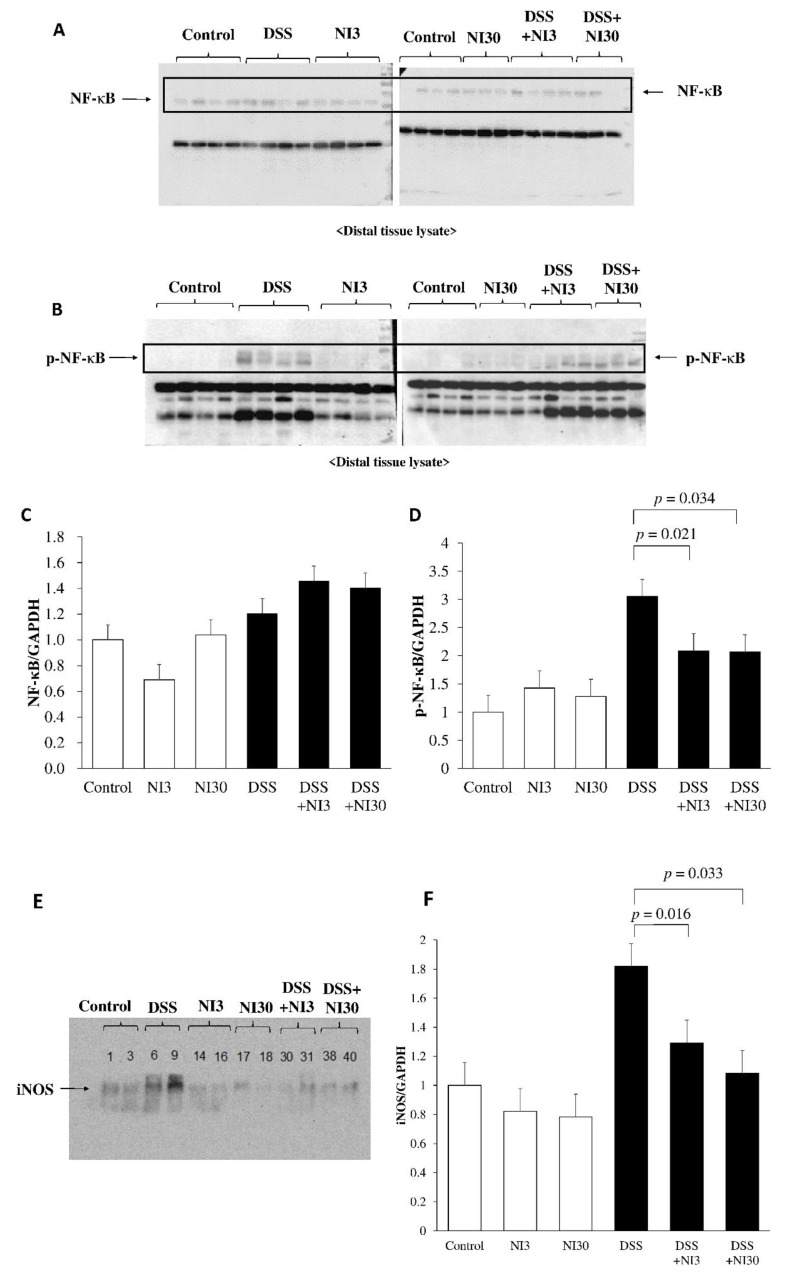
Effect of the necrosis inhibitor on NF-κB, p-NF-κB and iNOS levels. (**A**) Western blot analysis of NF-κB. (**B**) Western blot analysis of p-NF-κB. (**C**) NF-κB levels did not differ significantly. (**D**) P-NF-κB levels were significantly reduced in the NI-treated groups compared with those in the DSS group. (**E**) Western blot analysis of iNOS. (**F**) iNOS levels were significantly reduced in the NI-treated groups compared with those in the DSS group. (Abbreviations) DSS, dextran sulfate sodium; NI, necrosis inhibitor; iNOS, inducible nitric oxide synthase.

**Figure 6 pharmaceutics-15-00222-f006:**
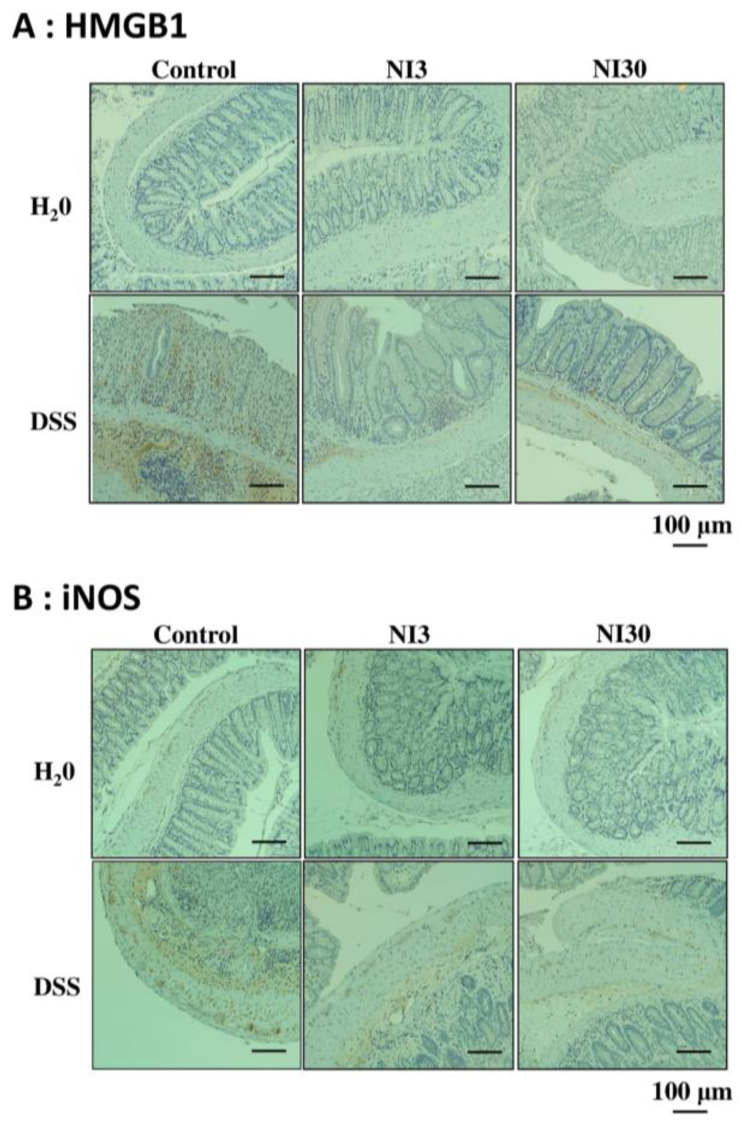
HMGB1 and iNOS immunohistochemical staining. HMGB1- and iNOS-positive cells are represented by the brown color. (**A**) HMGB1-positive cell counts were decreased in the NI3 and NI30 groups compared with those in the DSS group. (**B**) iNOS-positive cell counts were also decreased in the NI3 and NI30 groups (original magnification 100×). (Abbreviations) DSS, dextran sulfate sodium; NI, necrosis inhibitor, HMGB1, human mobility group box 1; iNOS, inducible nitric oxide synthase.

**Table 1 pharmaceutics-15-00222-t001:** Parameters of the disease activity index.

Feature	Score	Description
Weight loss (from baseline)	0	No weight loss or increase
	1	Weight loss of 1–10%
	2	Weight loss of 11–15%
	3	Weight loss of 16–20%
	4	Weight loss > 20%
Stool consistency	0	Well-formed pellets
	2	Semi-formed stool (no anal adherence)
	4	Liquid stool (anal adherence)
Bleeding	0	Absence
	2	Blood-tinged stool
	4	Gross bleeding

**Table 2 pharmaceutics-15-00222-t002:** Histological scores.

Feature	Score	Description
Inflammation severity	0	None
	1	Mild
	2	Moderate
	3	Severe
Inflammation extent	0	None
	1	Mucosa
	2	Mucosa and submucosa
	3	Transmural
Crypt damage	0	None
	1	Basal 1/3 damaged
	2	Basal 2/3 damaged
	3	Crypt loss and surface-epithelium retention
	4	Crypt and surface-epithelium loss
Percent involvement	0	None
	1	1–25%
	2	26–50%
	3	51–75%
	4	76–100%

## Data Availability

Data are available upon request due to restrictions. This experiment was conducted in collaboration with MitoImmune Therapeutics Inc. and data are available from the authors with the permission of MitoImmune Therapeutics Inc.

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
