# Peer review of "The Effect of Necrosis Inhibitor on Dextran Sulfate Sodium Induced Chronic Colitis Model in Mice"

_pharmaceutics, 2023, doi:10.3390/pharmaceutics15010222_

Round 1
Reviewer 1 Report
Authors from the manuscript titled “The Effect of Necrosis Inhibitor on Dextran Sulfate Sodium Induced Chronic Colitis Model in Mice” wanted to study the effects of NeroZ-7, a necrosis inhibitor, in a mice model of chronic colitis that was induced by dextrax sulfate sodium. Various methods were employed to aid in their assessment of the necrosis inhibitor on chronic colitis in mice. The manuscript has great intentions but there are too many English Language issues (almost every second sentence) with the manuscript to fairly judge the content and scientific robustness. Numerous statements are so vague it is difficult to understand what the authors are trying to say. It is strongly encouraged that the authors get a native English-speaking person to overhaul the manuscript so the full potential of this research can be valued. To add to this, there are a few factual statements that are incorrect and need to be improved. This may be due to the difficult language or an oversight, but in any manner needs to be changed.
For example: Cell death is divided into apoptosis and necrosis.
There are numerous cell death modalities
*Apoptosis does not change the cell structure nor
Yes it does as the cell shrinks and sends off apoptotic bodies.
*Necrosis occurs in way that is not controlled by external damage, etc.
In addition, the method section demands that the description is enough so anyone can repeat the experiments and get reproducible results. This is not the case in many areas of the methods: please improve.
Much more diligence is needed in the editing of the manuscript as the use of the template has useless information left in the manuscript. For example, This section may be divided by subheadings. It should provide a concise and precise description of the experimental results, their interpretation, as well as the experimental conclusions that can be drawn. Is not needed.
The overall result section is described but not every panel in the figure is which makes it hard for the reader to follow the logic of the results. More care in describing the results is needed for each result in order for the reader to be convinced that the data supports the author’s hypothesis. For example, only two sentences are used to describe figure 5. It would help to describe controls first, what the readers are supposed to see, and then what the data shows.
Discussion:
It is noteworthy to making sure that results are mentioned in the results and then discussed globally with comparison of others’ work. Please do this as there is results in the discussion that needs to be put into the results section, and a comparison is needed.
Provided a marked up pdf of highlights that have issues with the language.

Reviewer 2 Report
In this study, the authors examined the effect of the necrosis inhibitor NecroX-7 in the mouse model for induced chronic colitis. Overall, the manuscript was well written, the experiments were planned with great care and succinctly described while the results were of high quality, a testament to the hard work put in by the authors.
Just a few comments:
1) Figure 4B- the samples and protein standards appeared to be running on the same lane
2) Figure 5- the labels of the diagrams are unclear, the reader has to refer to the figure legend to find out which protein was examined. In addition, the description of the results for this set of results are scarce, with minimal information e.g. the iNOS are suppose to be in brown?
